# Does Video-Text Pretraining Help Open-Vocabulary Online Action Detection?

**Qingsong Zhao**[†1,2]    **Yi Wang**[†2]    **Jilan Xu**[†4,2]    **Yinan He**[†2]    **Zifan Song**[1]
**Limin Wang**[3,2]    **Yu Qiao**[2]    **Cairong Zhao**[*1]
[1]Tongji University    [2]Shanghai AI Laboratory    [3]Nanjing University    [4]Fudan University
{qingsongzhao, zhaocairong}@tongji.edu.cn
{wangyi, heyinan, qiaoyu}@pjlab.org.cn
lmwang.nju@gmail.com

## Abstract

Video understanding relies on accurate action detection for temporal analysis. However, existing mainstream methods have limitations in real-world applications due to their offline and closed-set evaluation approaches, as well as their dependence on manual annotations. To address these challenges and enable real-time action understanding in open-world scenarios, we propose OV-OAD, a zero-shot online action detector that leverages vision-language models and learns solely from text supervision. By introducing an object-centered decoder unit into a Transformer-based model, we aggregate frames with similar semantics using video-text correspondence. Extensive experiments on four action detection benchmarks demonstrate that OV-OAD outperforms other advanced zero-shot methods. Specifically, it achieves 37.5% mean average precision on THUMOS'14 and 73.8% calibrated average precision on TVSeries. This research establishes a robust baseline for zero-shot transfer in online action detection, enabling scalable solutions for open-world temporal understanding. The code will be available for download at https://github.com/OpenGVLab/OV-OAD.

## 1 Introduction

Action detection is a practical and demanding technique in intelligent video analysis, including anomaly detection [32] in surveillance and human-computer interaction [29] in embodied studies. Considering high variations in possible human behaviors with dynamic scenes, action detection is significantly challenging. In this regard, most action detection approaches go offline, involving the closed-set classification and localization of actions (a few predefined categories) within the long untrimmed videos. However, real-world applications concerning real-time understanding (e.g. surveillance) require estimating the action without accessing future frames. Further, closed-set discrimination limits the applicability of action detection, and it also asks for manually annotating all action categories, especially in complex scenarios such as a wide variety of actions or events in driving scenarios, which is both costly and time-consuming.

To address these challenges, we formulate online action detection in open-vocabulary and transfer popular vision-language models (VLM) to tackle this problem via only paired vision-text supervision for learning. A growing number of researchers have been investigating how to leverage the capabilities of powerful VLM to address specific novel visual tasks of interest. For instance, existing studies [21, 31, 1, 6, 25, 33] have explored the transfer of visual knowledge from VLM to a video understanding task to achieve zero-shot temporal action detection (ZS-TAD). Applying VLM to ZS-OAD is non-trivial. Plain solutions, as the ZS-TAD approaches mentioned earlier, involve partitioning a subset of

---

*corresponding author, †equal contribution

38th Conference on Neural Information Processing Systems (NeurIPS 2024).

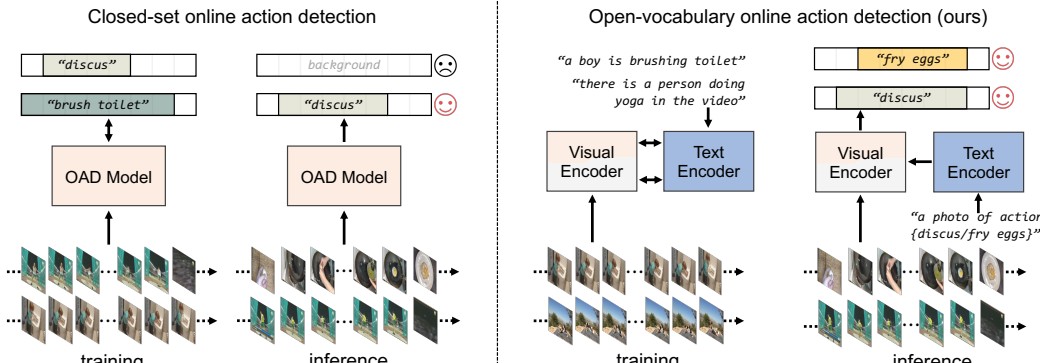

Figure 1: Overview of the online action detection. Models trained on closed-set actions (e.g., discus and brush toilet) are unable to detect the novel action class (e.g., fry eggs). We train a visual-text dual-encoder on web-collected video-text pairs without using frame-scale labels. It can discriminate arbitrary action classes.

base-to-novel category data from the downstream dataset and further fine-tuning the visual language model with a prompt-based technique to adapt it for novel tasks. However, this poses several issues. First, sliding-window frame sampling used in online action detection often leads to a high proportion of background frames, which contradicts the assumption of low background information for VLM training. Second, the OAD model fails to reach future frames during training, making it difficult to sample all category labels in the same batch. This is detrimental to the optimization of image-text contrast loss since the contrast loss favors the diversity of samples. We experimentally explored these hypotheses in Sec. 4.1.

Inspired by CLIP [34], given frames representation from a powerful VLM, we learn online motion detection models purely through text supervision, thus avoiding the use of fine-grained temporal annotations. To this end, we introduce the proposed object-centered decoder unit into the Transfomrer-based model, enabling automatic aggregation of frames with similar semantics with textual supervision exclusively. Fig. 1 illustrates the overall framework of our method. By employing contrast loss during training on extensive video-text pairs, we enable the model to be zero-shot transferred to different action detection vocabularies. Hence, we name our model Open-Vocabulary Online Action Detector (OV-OAD). We pre-train OV-OAD on the video-text datasets, and manual frame-level labels are not used whatsoever. We propose three proxy tasks including alignment of current frame-text embedding, background frame mask prediction, and alignment of multi-label video-text embedding for training. The first task enables the model to prioritize discriminative information from neighboring frames. The second task enables the successful detection of complex background frames in natural videos. The third task mitigates the impact of caption noise in web videos. Our model was evaluated on four action detection benchmarks without any fine-tuning, i.e., THUMOS'14 [19], EK100 [13], FineAction [27] and TVSeries [14] in a zero-shot manner. Extensive experiments demonstrate that our model outperforms other advanced zero-shot methods. The main contributions are summarized as follows:

- We investigate the critical problems of how to capitalize pre-trained visual language models for zero-shot online action detection in untrimmed videos.

- We introduce a novel video-text dual-encoder architecture, namely OV-OAD, to perform open-vocabulary online action detection. Experiments on the downstream datasets show that our model successfully learns clusters of similar video frames and transfers them to multiple action semantic vocabularies in a zero-shot manner.

- To our knowledge, our work is the first to explore zero-shot transfer from text supervision alone to the online action detection task without relying on any precise frame-scale labels. And we have established a robust baseline for this new setting.

## 2    Related Work

**Pretrained Vision-Language Model (VLM).**    Recently, the joint image-text learning paradigm [16] has been successfully scaled up by CLIP [34] and ALIGN [20] with the massive web data. After that, researchers have proposed many variations, including CLIP-Adapter [18], GLIP [24], and so on. One VLM's visual encoder can leverage textual descriptions to recognize objects or scenes in images when category-specific samples are unavailable. In video domains, similar ideas have been explored for action recognition (e.g., ActionCLIP [40], ViFi-CLIP [35]) and video understanding (e.g., CLIPBERT [22], EffPrompt [21]).

**Zero-Shot Temporal Action Detection**    Temporal action detection (TAD [36, 11, 48, 26]) is a video understanding task involving simultaneous recognition and localization of actions within an uncut video. Recently, efforts [21, 31, 1, 6, 25, 33] have utilized the pre-trained vision-language model to give the TAD models with the capability to recognize novel action classes. For example, EffPrompt [21] proposes a two-stage fine-tuning scheme for zero-shot temporal action detection (ZS-TAD) by incorporating task-specific prompt vectors. STALE [31] introduces a one-stage model to mitigate the error propagation problem encountered by EffPrompt by utilizing a parallel classification and localization design. T3AL [25] presents a training-free ZS-TAD that leverages an effective test-time augmentation strategy and external knowledge derived from generated subtitles. It is important to highlight that all those ZS-TAD methods adopt a Base-to-Novel fine-tuning approach, which involves dividing the dataset categories into training and inference subsets. Due to the strong diversity among the TAD datasets and the limitation of its scale size, it has been challenging to showcase the model's generalization capabilities. By contrast, we train our open-vocabulary online action detection model with large-scale video-text pairs only. During inference, the model does not require any additional fine-tuning to recognize arbitrary action classes.

**Onlne Action Detection.**    Contrary to offline motion detection, OAD does not predict action onset timing and cannot access future visuals. Arguably, OAD emphasizes real-time response and openness of recognition over the accuracy of action classification in practice. Existing researchers [17, 45, 15, 41, 46, 38, 4] often use closed-set datasets for training and testing, boosting recognition accuracy and speed on single datasets. For example, IDN [15] improves the discriminative representation of actions by selectively accumulating relevant information. OadTR [41] incorporates the fusion of current features and future frames for identifying ongoing actions. LSTR [46] captures contextual dependencies in videos, leading to improvements in action identification. E2E-LOAD [4] proposes an end-to-end framework that integrates a stream buffer between the spatial and spatiotemporal modeling. MAT [38] introduces a memory-anticipation-based pipeline to model the entire temporal structure of a video. In contrast, our work shifts the focus to enhancing the open recognition capability of OAD models. We aim to leverage readily available video-text pairs to zero-shot transfer visual knowledge into the OAD model, thereby improving the model's ability to handle unseen actions.

## 3    Methodology

Consider an untrimmed video $V$, we generate a clip sequence employing a sliding window of length $\tau$ on $V$ that moves frame by frame. On the $t$-th slide, we get a clip $V^t = \{V_{t-\tau}, \ldots, V_{t-1}, V_t\}$ where $V_t$ denotes $t$-th frame. Online action detection is to predict action probability $\widehat{y}_t$ in each frame $V_t$ using only past and current observations. We propose an open-vocabulary online action detection model (OV-OAD) for zero-shot online action detection with text supervision only. Our approach, illustrated in Fig. 2, consists of two primary components: a visual encoder and a text encoder. The visual encoder comprises a distant neighboring-frame transformer block and an action clustering one. We pre-train OV-OAD on a web-scale video-text dataset. In inference, we transfer the trained model to the zero-shot online action detection without any fine-tuning, as described in Sec. 3.3, and it can predict arbitrary action classes. We ignore the subscripts of individual images and text pairs for simplicity.

### 3.1    Architecture

#### 3.1.1    Visual Encoder
The visual encoder is composed of a distant neighboring-frame transformer block (with a light grey background in Fig. 2), and an action clustering block (with yellow background in Fig. 2) with an

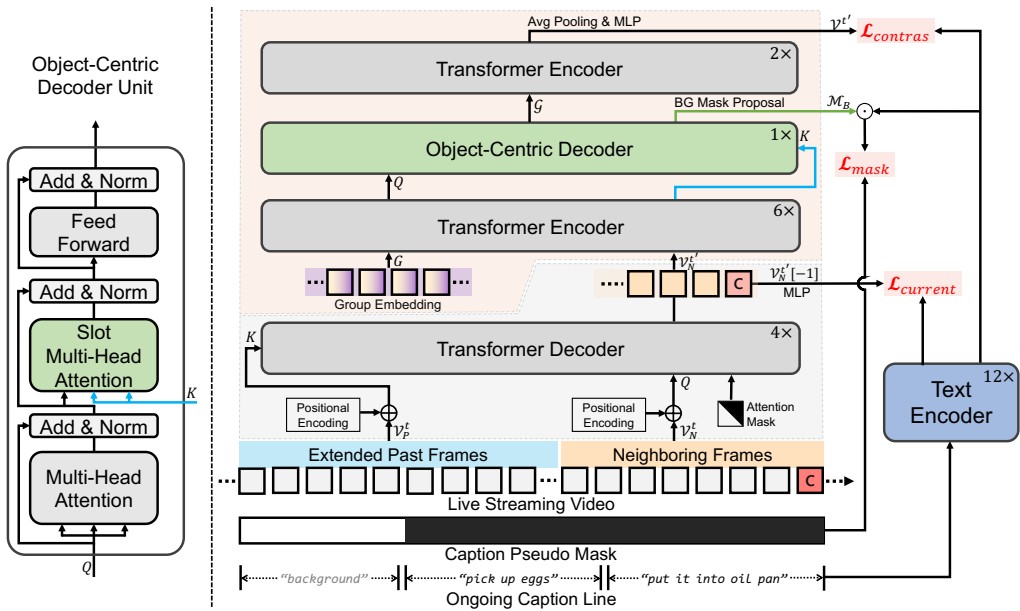

**Figure 2:** The illustration of our OV-OAD (best viewed in color), is formulated in a visual-text dual-encoder manner. Specifically, the visual encoder consists of a distant neighboring-frame transformer block ($\Psi_{DNTR}$, light grey backdrop) and an action clustering block ($\Psi_{AC}$, yellowish backdrop). The $\Psi_{DNTR}$ is built with Transformer Decoder units, which take the neighboring tokens and distant past tokens as inputs. The $\Psi_{AC}$ is built with our Object-Centric Decoder and vanilla Transformer Encoder units, which take the output tokens (orange squares) and learnable group embeddings (purple gradient squares) as inputs. During testing, the OV-OAD handles each incoming video snippet online, absent future context.

**object-centric decoder.** For a given video clip-text pair, denoted as $(\mathcal{V}^t, T)$, we initially divide the video into extended past frames $\mathcal{V}_P^t \in \mathbb{R}^{(\tau-n) \times d}$ and neighboring ones $\mathcal{V}_N^t \in \mathbb{R}^{n \times d}$. All frames $\mathcal{V}^t$ are processed in the Distant Neighboring-frame TRansformer block $\Psi_{DNTR}$, which comprises the transformer decoder unit [37]. Then the neighboring frames $\mathcal{V}_N^{t'}$ aggregated with the information of past frames are fed into the Action Clustering block $\Psi_{AC}$, along with $k$ learnable group embedding $(G \in \mathbb{R}^{k \times d})$, that aims to bind the neighboring frames into clusters. The outputs of the visual encoder are defined as:

$$(\mathcal{G}, \mathcal{V}^{t'}) := \Psi_{AC}(\left[G; \mathcal{V}_N^{t'}\right]) \circ \Psi_{DNTR}(\mathcal{V}_P^t, \mathcal{V}_N^t), \tag{1}$$

where the symbol $\circ$ represent a function composition, $\mathcal{G} \in \mathbb{R}^{k \times d}$ denotes the encoded group embeddings, while $\mathcal{V}^{t'} \in \mathbb{R}^{n \times d}$ refers to the output video frame tokens.

**Distant Neighboring-Frame Transformer.** It utilizes neighboring frames as queries to extract information from frames in the distant past. It is predominantly composed of four layers of standard transformer decoder units. Each frame in the video is augmented with 1-dimensional absolute positional encoding, independently applied to both past and neighboring frames. A directional attention mask is incorporated only for the neighboring frames, ensuring unidirectional information flow toward the current frame. Intuitively, the current frame embedding $\mathcal{V}_N^{t'}[-1]$ with more spatial information, will be aligned with the corresponding text as a raw video clip representation.

**Action Clustering.** The action clustering block namely $\Psi_{AC}$ assembles frames into groups and aligns the groups to human-understandable categories in a data-driven manner, only supervised by video-text pairs. It consists of three steps, i.e., the frame-to-group binding that assigns static frames with similar semantics to a group, the group-to-action mapping that computes the cosine similarity between the group embeddings with the action labels, and background mask proposing that predicts a set of binary masks by computing statistics from the frame-group similarity matrix $\mathcal{A}$.

In its training, we devise a binary prediction task to cluster frames into background and non-background as well as the grouping. The background mask $\mathcal{M}_B$ is predicted as:

$$\mathcal{A} := softmax(\frac{K@Q^\top}{\sqrt{d}}), \quad \mathcal{M}_B := \mathcal{A} \cdot \mathcal{G} \cdot \mathcal{T}^\top, \tag{2}$$

where $\mathcal{A} \in \mathbb{R}^{n \times k}$ is derived from the attentional weights in slot-attention computation. It denotes the likelihood of each video frame being assigned into $k$ learnable group embeddings. In Eq. 2, the group embeddings serve as the query $Q$, and frame tokens serve as keys and values. With the softmax operator normalizing over $k$, and the output of the slot-attention block is $\mathcal{A}^{\top}V$, which is of shape $k \times d$. The tensor $\mathcal{T} \in \mathbb{R}^{1 \times d}$ represents the $d$-dimensional embeddings of the video caption.

In its evaluation, similar to the computation of $\mathcal{M}_{\mathcal{B}}$, a video clip's action prediction score namely $\mathcal{P}_{AC} \in \mathbb{R}^{n \times C}$ can be calculated as follows:

$$\mathcal{P}_{AC} := \text{one-hot}(\arg \max_{k}(\mathcal{A})) \cdot \mathcal{G} \cdot \mathcal{T}_{val}^{\top}. \tag{3}$$

In contrast to Eq. 2, the frame-group similarity matrix $\mathcal{A}$ requires one-hot hard coding, while $\mathcal{T}_{val} \in \mathbb{R}^{C \times d}$ is encoded by the $C$ categories descriptions in the test set.

**Object-Centric Decoder Unit.** To group frames, we give an Object-Centric (OC) Decoder unit employing a slot attention mechanism [28] focused on the query object during cross-attention computation. It works similarly to the previously proposed GroupViT [43] and OVSegmentor [44] methods, which are specifically designed for semantic segmentation tasks.

As illustrated in Fig. 2, it requires two sets of inputs, i.e., target queries $\mathcal{G}$ consisting of a fixed number of $k$ encoded grouping embedding and $n$ input frame tokens $\mathcal{V}_{N}^{t'}$ to be queried, where $n$ can be a large number. Following a layer of multi-head self-attention, $\mathcal{G}$ is transformed to $\mathcal{G}'$. $\mathcal{G}'$ is then employed as a query in the second layer of multi-head slot-attention, while the frame tokens $\mathcal{V}_{N}^{t'}$ serve as the key and value. These two steps can be expressed as

$$\mathcal{G}' := softmax(\frac{\mathcal{G} \cdot \mathcal{G}^{\top}}{\sqrt{d}}) \cdot \mathcal{G}, \quad SlotAttn(\theta(\mathcal{G}'), \mathcal{V}_{N}^{t'}) := \left[ softmax(\frac{\mathcal{V}_{N}^{t'} \cdot \theta(\mathcal{G}')^{\top}}{\sqrt{d}}) \right]^{\top} \cdot \mathcal{V}_{N}^{t'}, \tag{4}$$

where $\theta : \mathbb{R}^{k \times d} \Rightarrow \mathbb{R}^{k \times d}$ denotes the dropout and norm operations.

### 3.1.2 Text Encoder

We adopt the pre-trained Text Transformer defined in CLIP [34] as the text encoder $\Psi_{T}$. To bridge the gap between image-text and video-text models, we utilize the Adaptformer technique [7], for lightweight transfer learning. We adopt the practice of setting up parallel adapters in each transformer block of $\Psi_{T}$. For the input text, we use the CLIP Tokenizer [34] to add the tokenizer start $[SOT]$ and tokenizer end $[EOT]$ at the beginning/end. The text embedding is computed as $\mathcal{T} = \Psi_{T}(T)$, where $T \in \mathbb{R}^{\cdot \times l}$ represents the tokenized caption with a length of $l$.

### 3.2 Video-text data for online action detection.

Web video-text datasets [8, 42] are abundant in data volume, but their captions are typically pre-labeled by image-text generative models [23, 9] and then filtered by humans semi-automatically. Consequently, these annotations are extremely noisy and include fantasy elements, despite being semantically rich and diverse. Additionally, we employ a sliding window approach to capture both the video frame and its corresponding text description during the live-streaming video. These factors introduce inherent bias in the visual and textual information of a sample pair. To achieve relatively accurate textual captions of the visual information in the video, we employ a multi-label contrast learning strategy. To generate multiple captions, we employ a linguistic analysis tool (e.g., the nltk toolkit [2]) when the raw labels are not enough. This tool extracts verb-object phrase structures from the given descriptions, which are then utilized as keywords to create additional captions, drawing on CLIP's prompting engineering. To handle redundant captions, we choose the text associated with the highest number of frames as the global descriptor, and the remaining captions are utilized to compute the multi-label contrast loss. In the absence of any tags, we utilize the keyword *"background"* for sentence construction and as the global descriptor.

### 3.3 Optimization and Inference

We train the model through three proxy tasks, i.e., video-text alignment, current frame-text matching, and background mask prediction. The total loss is: $\mathcal{L}_{\text{total}} = \mathcal{L}_{\text{contras}} + \alpha \mathcal{L}_{\text{current}} + \beta \mathcal{L}_{\text{mask}}$, where $\alpha, \beta$ are trade-off parameters controlling the relative weight of the above cost functions. Each used loss is detailed below.

**Current Frame-Text Alignment.** We adopt Image-Text Contrastive loss ($\mathcal{L}_{ITC}$) to learn whether the current frame image matches the caption. We denote the image embedding (from the distant neighboring-frame Transformer block) and the text embedding as $z^{I}$ and $z^{T}$, respectively. Both

embeddings are projected into a 512-dimensional joint feature space before calculating the matching loss. The current frame-text contrastive loss $\mathcal{L}_{current}$ is:

$$\mathcal{L}_{\text{current}} = \mathcal{L}_{\text{ITC}}(z^I, z^T) = -\frac{1}{2}\left(log\frac{\exp(z_i^I \cdot z_i^T/\tau)}{\sum_j^B \exp(z_i^I \cdot z_j^T/\tau)} + log\frac{\exp(z_i^T \cdot z_i^I/\tau)}{\sum_j^B \exp(z_i^T \cdot z_j^I/\tau)}\right), \quad (5)$$

where $\tau$ is a temperature parameter to scale the logits, and $B$ denotes the batch size.

**Multi-Label Video-Text Alignment.** We employ the multi-label video-text contrastive loss to align the visual and language representations for enhancing the textual representations of videos. The multi-label video-text matching loss $\mathcal{L}_{contras}$ is:

$$\mathcal{L}_{contras} = \mathcal{L}_{ITC}(z^V, z_0^T) - \frac{1}{2}\left(log\frac{\sum_m^M exp(z_i^I \cdot z_i^{T_m}/\tau)}{\sum_m^M \sum_j^B exp(z_i^I \cdot z_j^{T_m}/\tau)} + \frac{1}{M}\sum_{m=1}^M log\frac{exp(z_i^{T_m} \cdot z_i^I/\tau)}{\sum_j^B exp(z_i^{T_m} \cdot z_j^I/\tau)}\right),$$
$$(6)$$

where $z^V$ is computed as the average of the output $\mathcal{V}^{t'}$ from the visual encoder. $\{z^{T_0}, z^{T_1}, \ldots, z^{T_M}\}$ are text embeddings, constructed in Sec. 3.2. All embeddings are mapped into 256-dimensional vectors. Refer to Eq. 5, $z^{T_0}$ denotes a global descriptor selected from the multi-label captions.

**Background Mask Proposal.** Through video captions, we can roughly determine which frame chunks are background and which ones correspond to actions. This prior helps the action clustering block effectively group and bind the majority of background frames. To enhance the concatenation region between the predicted background mask $\mathcal{M}_B$ and the caption pseudo mask $\mathcal{M}_{GT}$, we employ a per-frame binary mask loss. Following Maskformer [10], we use dice loss [30] for our mask loss, i.e., $\mathcal{L}_{mask} = \mathcal{L}_{dice}(\mathcal{M}_B, \mathcal{M}_{GT})$. The binary mask $\mathcal{M}_{GT}$ is derived from the neighboring frames $\mathcal{V}_N^t$ of the input to the action clustering block. It is set to "1" for frames with a caption and "0" for frames without captions.

**Zero-Shot Online Inference** Similar to CLIP's zero-shot transfer [34], our distant neighboring-frame Transformer block can assign the current frame image to the semantic category with the highest image-text embedding similarity. During online inference, similar to Eq. 3, the action prediction score namely $\mathcal{P}_{DNTR} \in \mathbb{R}^{1 \times C}$ for the current frame of a video clip can be written as $\mathcal{P}_{DNTR} = \mathcal{V}_N^{t'}[-1] \cdot \mathcal{T}_{val}^\top$.

The action clustering block can **also** estimate frame action without fine-tuning. We calculate the similarity between the embedding of each frame token and the text embedding of the dataset. Then, we assign each frame token to the corresponding category with the highest similarity. This zero-shot transfer pipeline is depicted in Eq. 3. In summary, the final action prediction score $\widehat{y}_t \in \mathbb{R}^{1 \times C}$ for a video clip $V^t$ can be expressed as: $\widehat{y}_t = \mathcal{P}_{AC}[-1] + \alpha\mathcal{P}_{DNTR}$.

# 4 Experiments

Our run experiments on NVIDIA V100 $\times 8$ using Pytorch 1.11.0. During both training and inference, we resample the video raw frame rate (e.g., 24/30 FPS) to 4 FPS, and resize images to $224 \times 224$ [49, 46]. For feature extraction, we employ the CLIP model (ViT-L). Specifically, the visual encoder computes a series of image patch tokens along with one global token (aka, CLS token) for each frame, and we utilize the normalized CLS token as the output feature encoding. Unless otherwise specified, all parameters of the visual encoder in our OV-OAD model are initialized from scratch, while the text encoder is initialized with the CLIP, except for the additional Adapter parameters. We train our OV-OAD for 30 epochs with 2 warm-up epochs using the Adam optimizer with weight decay $5e^{-2}$. It uses a cosine schedule with a batch size of 256, and the initial learning rate is $1.6e^{-4}$.

**Pre-training Datasets.** We use the filtered InternVid-10M-FLT (aka, InternVid [42]) and the ActivityNet v1.3 (aka, ANet [3]) datasets for training, which are originally collected $\sim$4M and 14950 untrimmed video-caption pairs from the web, respectively. However, the videos within the InternVid dataset typically have longer durations compared to ANet, and the average percentage of foreground frames with annotations on these videos is only 27.4%. We sort the $\sim$4M videos in the InternVid dataset according to the number of caption annotations they contain and take the top 5000 videos (namely InternVid-5K) for training. For ANet, we utilize the prompting technique (following [35, 12]) to convert the short action tags into sentences. And we combined its training and test sets and utilized them collectively for training. Please see Appendix A.2 for complete dataset preparation.

**Benchmarks.** We follow previous works [47, 5, 4] and evaluate our model for the zero-shot online action detection on the validation splits of the THUMOS'14 [19], TVSeries [14], EPIC-Kitchens-100 (aka, EK100 [13]), and FineAction [27] datasets. THUMOS'14 and TVSeries datasets comprise 20

Table 1: Benchmark evaluation on THUMOS'14 and TVSeries. "IVid" denotes InternVid-5K.

| Methods | Arch | THUMOS'14 | TVSeries |
|---|---|---|---|
| | | mAP (%) | cAP (%) |
| CLIP-I | ViT/B | 28.0 | 67.7 |
| CLIP-I | ViT/L | 29.6 | 69.3 |
| CLIP-II | ViT/B | 29.1 | 69.5 |
| CLIP-II | ViT/L | 30.9 | 71.1 |
| CLIP-III | ViT/B | 29.7 | 71.6 |
| OV-OAD (IVid) | ViT/B | 33.2 | 73.8 |
| OV-OAD (ANet) | ViT/B | 37.5 | 73.2 |

Table 2: Benchmark evaluation on FineAction and EK100.

| Methods | Arch | FineAction | EK100 (Verb) |
|---|---|---|---|
| | | mAP (%) | cAP (%) |
| CLIP-I | ViT/B | 26.5 | 40.1 |
| CLIP-II | ViT/B | 27.8 | 39.9 |
| OV-OAD | ViT/B | 29.2 | 41.4 |

Table 3: Base-to-novel and fully-supervised evaluation on THUMOS'14 dataset.

| Train-Test Split | Methods | THUMOS'14 |
|---|---|---|
| | | mAP (%) |
| 100% Seen 0% Unseen | OadTR-D8 | 47.4 |
| | LSTR | 47.7 |
| | MAT-D48 | 48.2 |
| | CLIP-I$^\dagger$ | 28.0 |
| | OV-OAD$^\dagger$ | 37.5 |
| 75% Seen 25% Unseen | OadTR-D8 | 33.7 |
| | LSTR | 26.9 |
| | MAT-D48 | 25.5 |
| | CLIP-I$^\dagger$ | 38.6 |
| | OV-OAD$^\dagger$ | 44.6 |
| 50% Seen 50% Unseen | OadTR-D8 | 9.6 |
| | LSTR | 9.1 |
| | MAT-D48 | 7.9 |
| | CLIP-I$^\dagger$ | 28.6 |
| | OV-OAD$^\dagger$ | 35.9 |

and 30 foreground action categories, respectively. EK100 and FineAction datasets comprise 97 verb classes and 106 foreground action labels, respectively. An additional background class is considered for all datasets. Turn to the Appendix A.3 for dataset details.

We evaluate metrics for online motion detection based on previous studies [38, 46, 4]. Specifically, we applied per-frame mean average precision (mAP) on THUMOS'14 [19] and FineAction [27], and per-frame calibrated average precision (cAP) on TVSeries [14] and EK100 [13].

## 4.1 Comparison with Existing Methods

We conducted a comparison of the zero-shot online motion detection metrics between our method and other zero-shot baselines. We also explore the base-to-novel fine-tuning approach for zero-shot OAD and compare it to our OV-OAD model.

**Comparison with Zero-Shot Baselines.** We utilize visual language models with image zero-shot capabilities (i.e., CLIP) for comparison. The inference process of online action detection involves sliding frame-by-frame sampling on an untrimmed video and subsequently predicting the action class of the last frame (aka, the current frame). We can set the sliding length to 1 and classify the actions based on a single frame image to simplify the inference. To zero-shot transfer CLIP to online action detection, We first extract the features of the frame images using its visual encoder, and then, we compute the similarity between the visual features and the text embedding of the dataset action labels. We can perform several non-parametric processes on the visual embeddings, including: 1) Averaging the visual embeddings of the neighboring frames to obtain the visual feature of the current frame, named CLIP-II; 2) Non-parametric clustering of all sampled frames (e.g., K-means algorithm), followed by averaging the visual embedding of the group to which the current frame belongs, resulting in the visual feature, named CLIP-III; 3) Directly using the visual embedding of the current frame as the visual feature, dubbed as CLIP-I. Table 1 presents the experimental results, clearly demonstrating the superior performance of our OV-OAD over other non-parametric zero-shot methods. It is worth noting that enhancing the scale of the visual language model leads to a moderate increase in single-frame action prediction accuracy. However, this improvement is constrained, indicating that relying solely on robust image discrimination is insufficient for achieving high performance. Consequently, it is essential to incorporate temporal structure information into the learning process for effective online action recognition.

Furthermore, as depicted in Table 2, we delve into the zero-shot performance of OV-OAD on more demanding datasets. We evaluate OV-OAD's performance on the first-view shots dataset called EK100. This dataset comprises first-view shots and notably deviates from the ANet data distribution. We also assess the performance of OV-OAD on the large-scale dataset, FineAction, which includes ~4,000 uncut videos categorized into 106 action classes. The outcomes indicate that OV-OAD exhibits superior generalization capabilities in online action detection when contrasted with CLIP.

Table 4: Ablation study on the current frame-caption contrastive loss ($\mathcal{L}_{current}$) and background mask loss ($\mathcal{L}_{mask}$). The baseline only uses the multi-label video clip-text contrastive loss ($\mathcal{L}_{contras}$).

| $\mathcal{L}_{contras}$ | $\mathcal{L}_{current}$ | $\mathcal{L}_{mask}$ | mAP (%) |
|:---:|:---:|:---:|:---:|
| ✓ | | | 32.9 |
| ✓ | ✓ | | 36.3 |
| ✓ | | ✓ | 33.6 |
| ✓ | ✓ | ✓ | 37.5 |

Table 5: Results of different number of frame tokens about $\mathcal{V}_N^t$ and $\mathcal{V}_P^t$.

| $\mathcal{V}_P^t$ \ $\mathcal{V}_N^t$ | #4 | #8 |
|:---:|:---:|:---:|
| #16 | 36.1 | 37.1 |
| #24 | 36.7 | 37.5 |
| #28 | 37.1 | 36.9 |
| #32 | 35.9 | 36.5 |

Table 6: Results of different numbers of Transformer units for $\Psi_{AC}$ and $\Psi_{DNTR}$.

| $\Psi_{DNTR}$ | $\Psi_{AC}$ | mAP (%) |
|:---:|:---:|:---:|
| 0 | 9-3 | 31.9 |
| 0 | 5-3 | 33.2 |
| 0 | 6-6 | 33.3 |
| 4 | 6-2 | 37.5 |
| 4 | 6-0 | 36.9 |

Table 7: Results of different designs of the $\Psi_{DNTR}$ block. "TR" denotes Transformer. "OC" means Object-Centric. The penultimate row is our proposed OV-OAD design.

| Block $\Psi_{DNTR}$ | Clustering Unit | mAP (%) |
|:---|:---|:---:|
| n/a | OC Decoder | 33.3 |
| 4×TR Encoder | OC Decoder | 35.6 |
| 4×TR Encoder+1×Cross Attn | OC Decoder | 36.5 |
| 4×TR Decoder | OC Decoder | 37.5 |
| 4×TR Decoder | TR Decoder | 37.0 |

**Comparison with base-to-novel methods.** We compare base-to-novel fine-tuning methods and fully-supervised transfer to online action detection on the THUMOS'14 dataset. For base-to-novel generalization, we integrate three well-known Transformer-based online action detection models including OadTR [41], LSTR [46] and MAT [38] with a text encoder using the image-text contrastive loss. To ensure statistical significance, we adopted the random sampling setup and dataset partitioning method proposed by [21]. For our experiments, we employed two evaluation settings on the THU-MOS'14 dataset, i.e., training on 75% of the action categories and testing on the remaining 25%, and training on 50% of the categories while testing on the remaining 50%. We followed the experimental fine-tuning setup of [35], including the Adam optimizer, the learning rate $1e^{-3}$, and the Cosine decay function for training. For fully supervised transfer, we also train these models on 100% of the action categories with inputs of video frame features extracted by CLIP/ViT-B. We followed the training setup in [41, 46, 38] for the experiment, which remained consistent except for the different feature extractors. All experimental results are reported in Table 3, $^\dagger$ indicates that the model has not seen any categories and directly tests the performance of the unseen categories. The MAT-D48 indicates that MAT utilizes the ground truth of future frames (48 frames in 12 seconds) during training. We observe that the recent MAT method achieves better performance in the fully-supervised setting, but its performance is poor in the base-to-novel setting. One can find that our OV-OAD model outperforms the competition even without utilizing any training data. The results indicate that the base-to-novel fine-tuning method is not suitable for direct application to the zero-shot online action detection task. The limited availability of data may be a contributing factor to the unsuitability of the base-to-novel fine-tuning approach for online motion detection models.

## 4.2 Ablations

**Proxy Tasks.** We aim to validate the effectiveness of the three proxy tasks we introduced, namely current frame-text alignment, multi-label video-text alignment, and background mask proposal. As reported in Table 4, our baseline model employs the multi-label video-text contrastive loss $\mathcal{L}_{contras}$ only, upon incorporating the current frame-text matching loss $\mathcal{L}_{current}$, we observed a substantial improvement of 3.4% in mean Average Precision on the THUMOS'14 dataset. The performance improvement is due to the model's ability to capture spatio-temporal information from extended past frames. In addition, the prediction of the background frame mask also leads to improvements, and combining both results in optimal performance. This finding suggests that enhancing the model's capability to detect background frames is equally important.

**Number of Layers and Frames.** We first conduct an evaluation to assess the influence of inputting different numbers of neighboring and past frames on the model's performance. As depicted in Table 5, our OV-OAD model demonstrates flexibility with different frame choices, resulting in a maximum performance variation of 1.6%. Note that, the highest mean Average Precision is achieved when

utilizing $\mathcal{V}_N^t = 8$ and $\mathcal{V}_P^t = 24$. In addition, we investigate the effect of the number of network layers in different blocks on the performance. The results are presented in Table 6, we find that increasing the number of layers for the first Transformer Encoder of our action clustering block results in a notable decline in performance. Conversely, a small number of layers for the last Transformer Encoder proves to be sufficient.

**Distant Neighboring-Frame Transformer.** We further investigate the design of the proposed distant neighboring-frame transformer block $\Psi_{DNTR}$. Unless specified otherwise, we employ 2-second neighboring frames, 6-second distant past frames, and CLIP/ViT-B pre-trained features.

**a) Can we remove the $\Psi_{DNTR}$ block?** To implement this, we directly input all 8-second sampled video frames into our action clustering block, where the object-centered group module could easily cluster frame tokens that exhibit similarity. To ensure fairness, we set the number of Transformer layers in the action clustering block to be equal to the total Transformer layers in OV-OAD. As can be seen from Table 7 (row 4 vs. row 1), OV-OAD exceeds this baseline clearly. This also demonstrates the validity of our idea of applying neighboring frames to query spatio-temporal information from distant past frames.

**b) Can we remove the final Transformer encoder in $\Psi_{AC}$ block?** Experiments were conducted to analyze the impact of removing the final transformer encoder on the model's performance. The result presented in the Table 7 (row 5 vs. row 4) indicates a marginal performance decrease of around 0.6% upon removing the final transformer encoder. Additionally, this action results in a 15% reduction in certain training parameters, specifically in the visual encoder.

**c) Can $\Psi_{DNTR}$ block be learned efficiently using the Transformer encoder unit?** Here, we aim to explore whether the $\Psi_{DNTR}$ block can be learned efficiently using Transformer encoders only. To be specific, we combine the neighboring with distant past frames and feed them into a 4-layer $\Psi_{DNTR}$ block based on a standard Transformer encoder implementation. Table 7 (row 4 vs. row 2) illustrates that this baseline is clearly lower than our $\Psi_{DNTR}$ block constituted by the Transformer decoders. Furthermore, we introduced an additional layer of cross-attention after the 4-layer Transformer encoder to create a new baseline. We aim to assess the effectiveness of the "bottleneck" design of cross-attention within the $\Psi_{DNTR}$ block. Note that such an implementation also completes the process of querying discriminative information from distant past frames. Table 7 (row 4 vs. row 3) shows that the cross-attention design does exhibit effectiveness, but it falls short of achieving top performance.

**d) Ablation for the Action Clustering Block.** Here, we analyze the impact of an object-centric decoder compared to a standard Transformer decoder unit within the action clustering block. Both are designed to bind semantically similar static frames into a group embedding. Table 7 (row 5 vs. row 4) demonstrate that the object-centered decoder outperforms the standard transformer decoder in performance.

**On Adapting Text Encoder**   For our baseline approach, following the initialization of our OV-OAD's text encoder with the CLIP's text encoder weights, we release the weights to continue training. Then, we conduct experiments to explore the performance impact of two separate modifications: 1) fix the full backbone parameters and 2) incorporate the Adapter structure. The results are depicted in Table 8, one can see that leveraging the Adapter technique leads to a substantial improvement in performance. Moreover, to achieve the best results, it is necessary to release the backbone parameter and continue training.

### 4.2.1 Inference Speed.

We compared the model parameters and efficiency of our OV-OAD model with other methods on a single NVIDIA Tesla V100 GPU, given in Tab. 9. Note traditional supervised learning methods, the efficiency bottleneck of the system is primarily attributed to the optical flow computation and its feature extraction. Our OV-OAD eliminates the need for optical flow computation and the extraction of spatio-temporal features from the RGB image. The overall system achieves an impressive inference speed of 292.3 frames per second (FPS). The findings indicate that our model has the potential to be deployed on standard online video capture devices, enabling real-time action prediction capabilities. In particular, LSTR demands a larger number of input frames for optimal performance, using 520 seconds of video for inference on THUMOS'14, while our OV-OAD utilizes only 8 seconds. This means that LSTR requires 65 times more data than OV-OAD, which likely explains why our model's inference speed is six times faster. Furthermore, the primary speed bottleneck for LSTR is the extraction of optical flow (8.1 FPS), whereas for our model, it is the extraction of image features (292.3 FPS).

Table 8: Ablation study on Text Encoder.

| Pre-trained | Fixed | Adapter | mAP (%) |
|:---:|:---:|:---:|:---:|
| ✓ | | | 36.7 |
| ✓ | ✓ | | 35.8 |
| ✓ | | ✓ | **37.5** |
| ✓ | ✓ | ✓ | 35.5 |

Table 9: Efficiency comparison on parameter (M) and inference speed (FPS)

| Methods | #Param | Frames Per Second | | | |
|:---:|:---:|:---:|:---:|:---:|:---:|
| | | Optical Flow | RGB Feat | Flow Feat | Model |
| OadTR | 75.8M | | | | 110.0 |
| LSTR | 58.0M | 8.1 | 70.5 | 14.6 | 91.6 |
| MAT | 94.6M | | | | 72.6 |
| OV-OAD | 109.5M | - | 292.3 | - | 571.4 |

### 4.3 Limitations

End-to-end online motion detection systems require simultaneous learning of spatial and temporal structures for optimal results. Our OV-OAD model utilizes the CLIP's visual encoder to extract features from pure images. This would cause the network to focus too much on modeling foreground-centered spatial information at the expense of modeling spatio-temporal structure information. This does not fit the requirements of action recognition for visual representations since learned RGB features from a video commonly contain some of the temporal structural information, e.g., RGB features extracted by TSN [39] have some of the properties of optical flow features. Therefore extending OV-OAD to simultaneously model scenario and temporal information and enable action recognition with an open vocabulary remains a challenging problem.

**Failure Cases.** Our objective is to identify categories that exhibit poor recognition as well as those with high recognition rates. We provide a list of categories with the highest and lowest action recognition average precision in Table 10. Additionally, we present visual samples of these categories in Fig. 3. We find that the detection accuracy decreases in scenarios where the foreground of the action is relatively low, and multiple actions share similar backgrounds (e.g., "CliffDiving" and "CliffShot"). However, OV-OAD demonstrates better performance when the foreground or interacting objects are more distinct, as observed in cases like "CleanJerk" and "PoleVault". These findings suggest that future enhancements can focus on improving the recognition of fine-grained actions through joint modeling of spatio-temporal information.

Table 10: Action classes with the highest and lowest performance on THUMOS'14.

| Action Classes | CleanJerk | PoleVault | GolfSwing | CliffDiving | BaseballPitch | CricketBowling | Billiards | CricketShot |
|:---:|:---:|:---:|:---:|:---:|:---:|:---:|:---:|:---:|
| AP | 72.38 | 67.88 | 63.56 | 49.08 | 13.17 | 20.27 | 22.65 | 24.38 |

Figure 3: Failure recognition cases on THUMOS'14. We use the red box to indicate the location of the action that is taking place.

## 5 Conclusion

In our study, we take the initial step towards leveraging text learning for online action detection without explicit human supervision. Our findings demonstrate that by employing OV-OAD, reproductions acquired from large-scale video-text pairs, even with noise, can be successfully transferred to online action detection in a zero-shot manner. Moreover, we highlight that the conventional approach of base-to-novel fine-tuning does not yield favorable results on traditional online action detection test datasets. Instead, we illustrate that similar semantic frames can be directly clustered and transferred to downstream action detection datasets using abundant textual supervision.

## Acknowledgments

This work is supported by National Natural Science Fund of China (62076184, 62473286) in part by Shanghai Natural Science Foundation (22ZR1466700).

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

Table 11: Video-Text model evaluation on THUMOS'14.

| Methods | Arch | THUMOS'14 |
| --- | --- | --- |
| | | mAP (%) |
| ViCLIP-I | ViT/B | 23.0 |
| ViCLIP-II | ViT/B | 24.1 |
| ViCLIP-I | ViT/L | 25.7 |
| ViCLIP-II | ViT/L | 26.3 |
| CLIP-I | ViT/B | 28.0 |
| CLIP-II | ViT/B | 29.1 |
| OV-OAD | ViT/B | 37.5 |

Table 12: The impact of $P_{AC}$ for zero-shot performance on THUMOS'14.

| Weight of $P_{AC}$ | THUMOS'14 mAP (%) |
| --- | --- |
| 1.0 | 37.5 |
| 0.0 | 34.1 |

# A Appendix

## A.1 More Ablations

**Can we use the Video-Text model as a visual extractor?** Another promising visual extractor to consider is employing a video-text model, instead of an image-text model, utilizing a sliding window approach. We opted for ViCLIP [42], a straightforward video-text baseline model, to directly assess its performance for zero-shot online motion detection on the THUMOS'14 dataset. Following the pre-training settings of [42], we employed a sliding window that samples 8 frames for input into the visual encoder, aligning with the CLIP-II methodology. Moreover, we exclusively fed the last frame of the sliding window, aligning with the CLIP-I approach. The Table 11 showcases the results, indicating that employing the video-text pretraining model directly for zero-shot inference in online action detection results in unsatisfactory performance. Although ViCLIP can leverage information from the entire video during pretraining, its ability to capture only very brief video frames during online motion detection inference limits its performance significantly.

**Can we remove $P_{AC}$?** During inference, the prediction of the current frame comprises two components: the action clustering block ($P_{AC}$) and the distant neighboring frame Transformer block ($P_{DNTR}$), as indicated in Section 3.3. As illustrated in the Table 12, excluding the scores of $P_{AC}$ leads to significant performance drops.

## A.2 Dataset Preparation

**ANet.** ActivityNet v1.3 is an innovative and expansive benchmark dataset designed for human activity understanding in videos. It serves as a comprehensive resource to address the challenges of recognizing and analyzing a wide range of complex human activities relevant to everyday life. The main objective of ANet is to provide a diverse collection of video samples that cover a broad spectrum of human activities. Currently, the dataset offers samples from **203** distinct activity classes, ensuring a comprehensive representation of various actions and behaviors. On average, there are 137 untrimmed videos available per class, with each video containing an average of 1.41 instances of the corresponding activity. In total, the dataset comprises an impressive 849 hours of video content.

It is worth noting that ANet's annotation is not performed on live frames due to cost considerations. It only annotates the time points of meaningful actions present in a video, and not every semantic action is annotated. Since it filters the video before annotation, the duration of each video is relatively short and the proportion of background frames in the video is relatively low. After our statistics, its background frames account for **35.82**% of the total frames. Most of the researchers use ANet as a pre-training dataset for action recognition. Instead, we deal with it as a video text dataset. When

processing the ANet dataset, we employ the prompting project combined with the original action tags as keywords for sentence construction.

**InternVid-5K.** InternVid is a significant multimodal dataset that focuses on video-centric learning for multimodal understanding and generation tasks. It serves as a valuable resource for developing powerful and transferable video-text representations. The InternVid dataset is vast, comprising over 7 million videos with a cumulative duration of nearly 760,000 hours. Within this extensive video collection, there are 234 million video clips available, each accompanied by detailed descriptions that consist of a total of 4.1 billion words. The dataset's multimodal nature combines visual and textual information, enabling researchers to explore the relationship between videos and their accompanying descriptions.

The annotations provided by InternVid for this paper offer valuable information, but they pose challenges for our pre-training process. This is because video annotations generated using large models of image-text often contain text noise and inconsistencies. Additionally, these annotations tend to be concentrated in specific segments, lacking homogeneity. These features present difficulties for online action recognition tasks. To address these challenges, we conducted a thorough examination of the syntax in InternVid's text data and identified certain speech defects in the annotations. We filtered the training samples extensively to mitigate these issues. Initially, we selected 100,000 videos by sorting them based on the total number of annotated entries in each long-term video. Subsequently, we applied additional filtering based on the ratio of annotated frames to the total number of frames. Through this process, we selected 5000 samples of temporal text-pair data to form the InternVid-5K dataset. Depending on the acceptance of the paper, we plan to **release** the metafile of the modified dataset, which incorporates the aforementioned alterations, to further improve the quality and consistency of the annotations.

### A.3 Benchmarks

**THUMOS'14.** The THUMOS'14 (THUMOS 2014) dataset is a significant video dataset widely used for action detection and recognition tasks. In the THUMOS'14 dataset, there are 220 videos in the validation set and 212 videos in the testing set that have been annotated with temporal boundaries. These annotations provide precise information about the start and end times of specific actions within the videos. With its extensive video collection, class diversity, and temporal annotations, the THUMOS'14 dataset serves as a valuable resource for advancing state-of-the-art in action detection research.

**TVSeries.** The TVSeries Dataset is a comprehensive and realistic large-scale dataset specifically designed for action detection tasks. It encompasses a total of 16 hours of video content extracted from six recent TV series. The dataset includes a diverse range of scenes and contexts, offering a representative sample of real-world action scenarios. Within the TVSeries Dataset, there are thirty distinct action classes defined, covering a wide spectrum of human activities. Each action instance in the dataset is meticulously annotated with precise start and end times, providing valuable temporal information for action detection algorithms.

**FineAction.** FineAction comprises 103,000 temporal instances across 106 action categories, annotated within 17,000 untrimmed videos. The dataset offers new opportunities and challenges for online action detection, characterized by finely defined action classes, diverse attributes, multiple instance annotations, and concurrent actions from various classes. With around 4,000 uncut videos categorized into 106 classes like "Household Activities", "Personal Care", "Socializing", "Relaxing", "Sports", and "Exercise", FineAction sets the stage for innovative research.

**EPIC-KITCHENS-100.** EPIC-KITCHENS-100 (EK100) includes first-person perspective shots and significantly differs from the ANet data distribution. EK100 contains 100 hours of video, 20 million frames, and 90,000 action segments across 45 environments, with narrations mapped to 97 verb classes and 300 noun classes. This dataset addresses various challenges such as action recognition, detection, and anticipation, offering a platform to assess model generalization across time and diverse contexts.

## A.4  Explain the Performance Differences on THUMOS'14 and TVSeries.

**The case of pre-training with ANet.**    The variance in improvements between THUMOS'14 and the TV series can be attributed to the resemblances in data distribution across these two datasets and our utilization of ANet. The substantial boost of OV-OAD on THUMOS'14 can be attributed to ANet encompassing a broader array of action categories. By employing a method of close comparison in natural language, we identified 8 similar action phrases within THUMOS'14's 20 action categories, constituting 40% of its overall categories. In the TV series dataset, we identified 9 similar action categories out of 30, equating to 30% of its total categories.

**Using different pre-training datasets.**    Similarly, OV-OAD achieves better performance on THU-MOS'14 due to the fact that ANet covers a wider range of action categories than IVid. Using the same natural language tools, we compared the coverage ratios of ANet and IVid for THUMOS categories, which stood at 40% and 15%, respectively. To elaborate, when contrasting the IVid and THUMOS datasets, we considered 500 high-frequency verbs as the action categories for IVid. Then, we pinpointed 3 categories from them that were similar to THUMOS'14.

