# OpenReview forum: "Does Video-Text Pretraining Help Open-Vocabulary Online Action Detection?"
_NeurIPS.cc/2024/Conference — NeurIPS 2024 poster_

### Official Review · Reviewer_wz2n · 2024-07-03

**Soundness:** 3
**Presentation:** 3
**Contribution:** 3
**Rating:** 6
**Confidence:** 4

**Summary:**

This paper explores the problem of open vocabulary video action detection in an online setting, where the action must be detected immediately once it appears in the video stream, vs. the more common offline setting that allows examining the entire video, past and future.

The authors propose a model with two main components: a transformer decoder that cross-attends between recent and past video frames, and an action clustering block that uses slot attention to group related frames and then classify them by action.

The model is trained on combination of three tasks/losses: contrastive image-text loss between the current frame's visual embedding and text embedding (vs. text embeddings from other clips in the batch), multi-label contrastive video-text loss (for text clips described by multiple text labels), and mask loss for identifying which frames come from the background (no action present).

The model demonstrates improved performance over CLIP baselines in the novel open vocabulary online action detection setting.

**Strengths:**

The problem setting introduced in this paper, online-streaming action recognition with an open vocabulary, is very realistic for modeling many practical real-world scenarios, e.g. home security cameras. Further, the model is impressively fast at inference (292 fps), making it quite reasonable to practically apply in this setting.

The paper includes an expansive, robust set of ablation experiments to validate the numerous model design choices.

I appreciate that the paper calls out joint modeling of spatio-temporal information as a limitation and backs this up with an analysis of the model's failure cases.

**Weaknesses:**

Line 61 says that "our model successfully learns clusters of similar video frames", and I see that indirectly based on the fact that there's a clustering module that leads to improved performance. However such a claim would be stronger if supported by evidence or examples of frames being correctly clustering based on action semantics.

The Object-Centric Decoder unit seems to be confusingly-named, since it is employed "to group frames" (line 149), rather than to focus on specific objects appearing within a frame.

In Table 1, in the proposed setting, the model is compared only against CLIP variants. Another baseline worth exploring would be video-text models (as opposed to image-text) applied via a sliding window.

There are a few parts of the paper that could be explained more quickly, in my opinion. (See the questions below.)

**Questions:**

Do you see a difference between "open vocabulary" and "zero shot" action recognition? The terms seem to be used synonymously in this paper, e.g. lines 106-107.

Related, how much overlap is there between the textual descriptions in ActivityNet and InternVid  with the category label sets of THUMOS'14 and TVSeries?

Can you explain "the assumption of low background information for VLM" from line 36 in more detail?

Can you please clarify "fails to reach future frames during training" from line 37?

What precisely is the circle operator in equation (1)?

On Line 175, how do you know "when the raw labels are not enough"?

In Table 2, OVO_{ad}^{dagger} refers to OV-OAD, correct?

**Limitations:**

yes

---

> ### Author Rebuttal · Authors · 2024-08-07
>
> Thanks for your valuable comments. Below are our responses to your concerns:
> ### Q1: However such a claim would be stronger if supported by evidence or examples of frames being correctly clustering based on action semantics.
> Here, we give evidence both quantitatively and qualitatively.
> - Firstly, during inference, the prediction of the current frame comprises two components: the action clustering block ($P_{AC}$) and the distant neighboring frame Transformer block ($P_{DNTR}$), as indicated in Line 215. As illustrated in the table below, excluding the scores of $P_{AC}$ leads to significant performance drops.
> | Weight of $P_{AC}$ | THUMOS'14 (mAP%) |
> |:-----|:----:|
> | 1.0 |37.5|
> | 0.0 |34.1|
>
> - Secondly,  in the final version, we will visualize the attention matrix $A$ in the object-centric decoder. Following the computation of Eq. 3, we can obtain a representation of the grouping of the input video frames.
>
>
> ### Q2: The object-centric decoder unit seems to be confusingly-named, since it is employed "to group frames" (line 149), rather than to focus on specific objects appearing within a frame.
> We will clarify that in the object-centric decoder, the term "objects" refers to video frames.
>
> ### Q3: Another baseline worth exploring would be video-text models (as opposed to image-text) applied via a sliding window.
> We opted for ViCLIP [40], a straightforward video-text baseline model, to directly assess its performance for zero-shot online motion detection on the THUMOS'14 dataset. Following the pre-training settings of [40], we employed a sliding window that samples 8 frames for input into the visual encoder, aligning with the CLIP-II methodology. Moreover, we exclusively fed the last frame of the sliding window, aligning with the CLIP-I approach. The results presented in the table below demonstrate that utilizing the video-text pretraining model directly for zero-shot inference in online action detection yields unsatisfactory performance.
>
> ViCLIP can access information from the entire video during pre-training; however, it can only capture very brief video frames during online motion detection inference. This limitation constrains its performance.
> | Methods | Arch | THUMOS'14 (mAP%) |
> |:-----|:----:|:----:|
> | ViCLIP-I  | ViT/B  |23.0|
> | ViCLIP-II  | ViT/B |24.1|
> | ViCLIP-I  | ViT/L | 25.7|
> | ViCLIP-II  | ViT/L |26.3|
> | CLIP-I     | ViT/B | 28.0|
> | CLIP-II    | ViT/B | 29.1|
> | OV-OAD | ViT/B | **37.5**|
>
> ### Q4: Do you see a difference between "open vocabulary" and "zero shot" action recognition?
> The disparity between "open-vocabulary" and "zero-shot" action recognition lies in the fact that open-vocabulary enables the recognition of any category, whereas "zero-shot" ensures that the testing categories were not encountered during training [a].
>
> In Line 106-107, we label our approach as "open-vocabulary" since our model is designed to ideally facilitate the online detection of any actions with the assistance of a vision-language model. In this context, "zero-shot" signifies that our method can be directly applied for inference on downstream datasets without the need for additional fine-tuning.
>
> ### Q5: How much overlap between the textual descriptions in ActivityNet and InternVid with the category label sets of THUMOS'14 and TVSeries?
> By utilizing the method of proximate comparison in natural language, we have pinpointed 8 analogous action phrases within the 20 action categories of THUMOS'14, encompassing 40% of its total categories. Likewise, we have identified 9 similar action categories out of the 30 in TVSeries, representing 30% of its total categories. For more detailed information, please refer to the response To-All-Reviewers-Q3.
>
> ### Q6: Can you explain "the assumption of low background information for VLM" from line 36?
> VLM are primarily designed for classification tasks rather than detection tasks. The optimization process of these models typically operates under the assumption that the background information (such as irrelevant environmental content) in the training data is less significant compared to the foreground content (task-relevant objects or regions). Essentially, these models are trained with the notion that each sample's main focus is on the foreground content, with less emphasis on the background, likely because classification tasks typically involve identifying the main objects in an image without detailed scene analysis. Consequently, VLMs may not be well-suited for detection tasks that necessitate precise differentiation between foreground and background elements. A similar perspective is echoed in STALE [29].
>
> ### Q7: Can you clarify "fails to reach future frames during training" from line 37?
> Given an untrimmed long video, the temporal action detection model can see the features of all frames, while the OAD model can only see the current frame and past frames in each iteration.
>
> ### Q8:  What precisely is the circle operator in equation (1)?
> In this paper, the symbol "◦" represent function composition. If $Ψ_{AC}$ and $Ψ_{DNTR}$ are two functions, then $Ψ_{AC}◦Ψ_{DNTR}$ denotes the composition of them, which is defined as $Ψ_{AC}◦Ψ_{DNTR}=Ψ_{AC}(Ψ_{DNTR}(x))$.
>
> ### Q9:  On Line 175, how do you know "when the raw labels are not enough"?
> In this context, "raw labels" refer to the captions associated with the selected video clips. Descriptions in large-scale video datasets often lack density. The OAD model utilizes a sliding window to extract multiple frames across the video, easily picking up segments with sparse or minimal captions.
>
> ### Q10:  In Table 2, VO_{ad}^{dagger} refers to OV-OAD, correct?
> That's a typo. OVO_{ad}^{dagger} means OV-OAD.
>
> ### Reference
> [a] Hyun, Jeongseok, et al. "Exploring Scalability of Self-Training for Open-Vocabulary Temporal Action Localization." arXiv preprint arXiv:2407.07024 (2024).

---

> > ### Comment · Reviewer_wz2n · 2024-08-10
> >
> > Thank you for the additional clarification, and the analysis of the performance of a video-text baseline. I appreciate the additional inclusion of benchmarks vs. more recent techniques, and on more datasets. I believe all of my concerns were addressed.

---

> > > ### Author Response · Authors · 2024-08-13
> > > **Please let us know if your concerns have been addressed**
> > >
> > > Dear Reviewer wz2n,
> > >
> > > We wish to express our gratitude for your extensive review and positive comments. **Your comments are invaluable to us, and we are fully committed to thoughtfully incorporating your insights to enhance our paper**. As the discussion phase is nearing its end, we are warmly concerned whether our rebuttal addresses your concerns.
> > >
> > > **It would be appreciated if you could raise your score on our paper if we address all your concerns.** We thank you again for your effort in reviewing our paper.
> > >
> > > Best regards,
> > >
> > > Authors of Paper #1178

---

### Official Review · Reviewer_Boy7 · 2024-07-09

**Soundness:** 3
**Presentation:** 3
**Contribution:** 2
**Rating:** 6
**Confidence:** 3

**Summary:**

This paper addresses a challenging setting in video understanding: open-vocabulary online action detection. It leverages pre-trained visual language models with a proposed dual-encoder architecture to achieve successful zero-shot detection.

**Strengths:**

1. The proposed method follows a visual-text dual encoder approach and applies it novelly to zero-shot online temporal action detection, achieving state-of-the-art performance.
2. Detailed ablation studies and analyses are included. This paper provides comprehensive ablations regarding the architecture design, loss design, and efficiency analysis, demonstrating the advantages of the proposed method over previous approaches.
3. Clear writing. The paper is well-organized and easy to follow.

**Weaknesses:**

1. The choice of VLM. The method uses CLIP as the visual-text encoder. However, as discussed in the limitations section, CLIP is an image-based understanding model and lacks the capability to capture temporal context. I wonder if other VLMs, such as ActionCLIP, could mitigate these drawbacks for the proposed task.
2. The comparison of the proposed method with previous work, such as OadTR, seems to use different visual encoders/features. Can the authors explain the fairness of such a comparison? Besides, the author should also compare the method with more recent methods, since OadTR is not state-of-the-art in recent works.

**Questions:**

Please see the weaknesses section.

**Limitations:**

Limitations have been well discussed by the authors in the paper.

---

> ### Author Rebuttal · Authors · 2024-08-07
>
> Thanks for your valuable comments. Below are our responses to your concerns:
> ### Q1:  I wonder if other VLMs, such as ActionCLIP, could mitigate these drawbacks for the proposed task.
> Thank you for the suggestion. We plan to enhance our model's performance in future work by incorporating improved feature extractors like ActionCLIP [38].
>
> ### Q2: The comparison of OV-OAD with OadTR, seems to use different visual encoders/features.
> Thank you for pointing out the lack of description here. In line 276, we noted that both the fully supervised methods and our OV-OAD utilize the same feature extractor. Specifically, both the base-to-novel fine-tuning methods and the fully supervised methods employ the same feature extractor, namely CLIP's visual encoder ViT/B. In the final version, we will emphasize this statement at the beginning of the paragraph.
>
> ### Q3: The author should also compare the method with more recent methods.
> Please refer to the response To-All-Reviewers-Q1 for detials.

---

> > ### Comment · Reviewer_Boy7 · 2024-08-11
> >
> > I appreciate the author's effort during the rebuttal. The author has clarified most of my concerns. Considering weakness 1 and other reviewers' opinions, I will keep my score as borderline accept.

---

> > > ### Author Response · Authors · 2024-08-13
> > > **Response to Comments**
> > >
> > > Thank you for taking the time to review our response and for providing valuable feedback. We are pleased to hear that the majority of your concerns have been addressed.
> > >
> > > Indeed, your question Q1 has inspired us significantly. In the limitation analysis of the initial draft of our paper, we made a similar intuitive assumption that better vision-language models (VLMs) could further enhance the performance of zero-shot online action detection. We are actively exploring the use of CLIP variants of VLMs (e.g., ActionCLIP) for feature extraction of video frames. However, due to resource constraints within our group, we are currently in the process of extracting features from the **14,950** videos in the ANet dataset using **ActionCLIP ViT-B/16**, which involves frame-by-frame extraction and saving the features to a storage device. We have made significant progress, with approximately **80%** of the extraction complete. We are committed to addressing all your concerns and will incorporate the aforementioned experiments and analyses into the final version of the paper.
> > >
> > > Moving forward, our primary direction for future research is to improve the feature extraction backbone, building on the insights gained from this work.

---

> > > > ### Author Response · Authors · 2024-08-13
> > > > **Please let us know if your concerns have been addressed**
> > > >
> > > > Dear Reviewer Boy7,
> > > >
> > > > We wish to express our gratitude for your extensive review and positive feedback. Your feedback is invaluable to us, and **we are fully committed to thoughtfully incorporating your insights to enhance our paper.** As the discussion phase is nearing its end, we would be grateful to hear your feedback and wondered if there are any other concerns we could address, especially as mentioned in your feedback regarding weakness 1 and other reviewers' opinions.
> > > >
> > > > **It would be appreciated if you could raise your score on our paper if we address your concerns.** We thank you again for your effort in reviewing our paper.
> > > >
> > > > Best regards,
> > > >
> > > > Authors of Paper #1178

---

> > > > > ### Author Response · Authors · 2024-08-14
> > > > > **Official Comment by Authors**
> > > > >
> > > > > Dear Reviewer Boy7,
> > > > >
> > > > > We appreciate your increased score! **Your insights have been incredibly helpful, and we are excited to incorporate the changes based on your suggestions into our paper.**
> > > > >
> > > > > Once again, thank you sincerely for your support and valuable feedback!
> > > > >
> > > > > Best regards,
> > > > >
> > > > > Authors of Paper #1178

---

### Official Review · Reviewer_zhJf · 2024-07-11

**Soundness:** 3
**Presentation:** 3
**Contribution:** 3
**Rating:** 6
**Confidence:** 5

**Summary:**

The paper introduces OV-OAD, a novel zero-shot online action detection system leveraging vision-language models for open-world temporal understanding. The authors propose a Transformer-based model with an object-centered decoder unit, trained solely on video-text pairs without manual frame-level annotations. The system is evaluated on THUMOS'14 and TVSeries benchmarks, demonstrating superior performance over existing zero-shot methods.

**Strengths:**

1. Innovation: The paper introduces a significant advancement in action detection by proposing a zero-shot online detection method that does not rely on manual annotations.
2. Technical Depth: The proposed model incorporates a novel object-centered decoder unit within the Transformer framework, which is a sophisticated approach to frame aggregation.
3. Experimental Validation:  The paper provides extensive experiments on two benchmarks, demonstrating the effectiveness of the proposed method.

**Weaknesses:**

1. The OadTR model was proposed in 2021, and the authors should consider using more recent OAD models such as MAT, MiniROAD, etc., as comparative benchmarks to enhance the timeliness and competitiveness of their model.
2. The authors have employed a multitude of complex modules for Zero-shot experiments on two datasets, with improvements on the TVSeries dataset being relatively limited compared to THUMOS14. This raises doubts about whether the proposed model is robust enough to serve as a standard baseline for future research.
3. As a pioneering work, the authors have experimented on typical action datasets like THUMOS14 and TVSeries. However, it would be beneficial to also test on atypical human action datasets, such as creating a benchmark for the HDD dataset.
4. In the action clustering block, which serves as the query and which as the key and value between the input Group Embedding and the output tokens? According to Figure 2, it seems that the Group Embedding is the query, and the latter is the key and value, but why is the dimension of the attention weight matrix $A$ $n \times k$ instead of $k \times n$? (line 142)
5. Line 154 mentions that the number of neighboring frames n can be a large value, and Table 4 shows the highest $n$ up to 8. Can the authors present experimental results for higher values of $n$?
6. Could the authors further explain how the model achieves ultra-high inference speed with a larger scale of parameters? For example, the parameter amount is twice that of LSTR, but the inference speed is six times as fast.
7. The paper lacks explanations for some operations, such as the operator "$\circ$" in Equation 1. This may lead to misunderstandings, mistaking it for a Hadamard product rather than a composite function.
8. It is recommended that the authors add visible variables in the figures used to illustrate the method (such as Figure 2) and improve the explanation of different parts of the chart to enhance readability.

**Questions:**

1. The OadTR model was proposed in 2021, and the authors should consider using more recent OAD models such as MAT, MiniROAD, etc., as comparative benchmarks to enhance the timeliness and competitiveness of their model.
2. The authors have employed a multitude of complex modules for Zero-shot experiments on two datasets, with improvements on the TVSeries dataset being relatively limited compared to THUMOS14. This raises doubts about whether the proposed model is robust enough to serve as a standard baseline for future research.
3. As a pioneering work, the authors have experimented on typical action datasets like THUMOS14 and TVSeries. However, it would be beneficial to also test on atypical human action datasets, such as creating a benchmark for the HDD dataset.
4. In the action clustering block, which serves as the query and which as the key and value between the input Group Embedding and the output tokens? According to Figure 2, it seems that the Group Embedding is the query, and the latter is the key and value, but why is the dimension of the attention weight matrix $A$ $n \times k$ instead of $k \times n$? (line 142)
5. Line 154 mentions that the number of neighboring frames n can be a large value, and Table 4 shows the highest $n$ up to 8. Can the authors present experimental results for higher values of $n$?
6. Could the authors further explain how the model achieves ultra-high inference speed with a larger scale of parameters? For example, the parameter amount is twice that of LSTR, but the inference speed is six times as fast.
7. The paper lacks explanations for some operations, such as the operator "$\circ$" in Equation 1. This may lead to misunderstandings, mistaking it for a Hadamard product rather than a composite function.
8. It is recommended that the authors add visible variables in the figures used to illustrate the method (such as Figure 2) and improve the explanation of different parts of the chart to enhance readability.

**Limitations:**

In summary, the authors have examined the OAD (Online Action Detection) from an interesting perspective and proposed a novel Zero-shot model, which appears to have profound application value compared to traditional OAD methods. However, additional experiments may be necessary to substantiate the claims fully.

---

> ### Author Rebuttal · Authors · 2024-08-07
>
> Thanks for your valuable comments. Below are our responses to your concerns:
> ### Q1: The authors should consider using more recent OAD models.
> Please refer to the response To-All-Reviewers-Q1 for detials.
>
> ### Q2: The improvements on the TVSeries dataset being relatively limited compared to THUMOS'14.
> We believe that there are two reasons why OVOAD's performance in THUMOS'14 outperforms the improvements in the TVSeries dataset.
> - First, all categories of ANet have different coverage ratios of **40%** and **30%** for the action categories of THUMOS14 and TVSeries, respectively. Similar text descriptions encountered by the text encoder during pre-training improve the accuracy of visual-text matching. Please refer to the response To-All-Reviewers-Q3 for specific calculations.
> - Second, the evaluation metrics are calculated differently for the THUMOS'14 and TVSeries datasets, the former used mAP as a metric while the latter used cAP as a metric.
>
> ### Q3: It would be beneficial to test on atypical human action datasets, such as creating a benchmark for the HDD dataset.
> Thank you for your advice. We have validated the zero-shot performance of OV-OAD on EK100 with non-trivial improvement (1.3% cAP). Please refer to the response To-All-Reviewers-Q2 for detials. In the final version, we will evaluate our method on those challenging online motion detection datasets, e.g., FineAction and the mentioned HDD.
>
> HDD [a] is a multimodal dataset in a driving scenario. In previous studies, e.g., OadTR [39] and Colar [45], non-visual sensors data are generally used as inputs to the model. The non-visual sensors are sourced from the vehicle's controller area network bus, encompassing critical metrics like car speed, accelerator and braking pedal positions, yaw rate, steering wheel angle, and the rotation speed of the steering wheel. In contrast, OV-OAD has only a visual feature extractor and does not directly process sensor data. Directly evaluating our current model on HDD could lead to unsatisfying performance due to the huge data distribution gap between our training (RGB videos for daily actions) and HDD.
>
> ### Q4: The group embedding is the query, and the frame tokens is the key and value, but why is the dimension of the attention weight matrix $A \in n×k$ instead of $k×n$?
> In the action clustering block, the group embeddings serve as the query, and frame tokens serve as keys and values. As shown in Eq.(4), we define  $A = softmax(\frac{K@Q^{\top}}{\sqrt{d}}), A \in \mathbb{R}^{n \times k}$, with the softmax operator normalizing over $k$, and the output of the slot-attention block is $A^{\top}V$, which is of shape $k×d$.  We will clarify the definition of $A$ in the final version.
>
> ### Q5: Line 154 mentions that the number of neighboring frames $n$ can be a large value, and Table 4 shows the highest $n$ up to 8. Can the authors present experimental results for higher values of $n$?
> Training our model with a larger $n$ than 8 (e.g. 12) needs V100×8 GPUs for ~5 days on ANet. We will add this study in the final version. We state that $n$ can be a large number relative to the number of group embeddings $k$. Generally, $n$ should be more than twice $k$. We will make these statements clear.
>
> ### Q6: The parameter amount is twice that of LSTR, but the inference speed is six times as fast.
> The inference of LSTR is much slower than ours due to two main reasons:
> - It requires more input frames than our method.
> - It relies on optic flow extraction, which is a slow process.
>
> In particular, LSTR demands a larger number of input frames for optimal performance, using 520 seconds of video for inference on THUMOS'14, while our OV-OAD utilizes only 8 seconds. This means that LSTR requires **65** times more data than OV-OAD, which likely explains why our model's inference speed is six times faster. Furthermore, the primary speed bottleneck for LSTR is the extraction of optical flow (8.1 FPS), whereas for our model, it is the extraction of image features (292.3 FPS).
>
> ### Q7: The paper lacks explanations for some operations, such as the operator "◦" in Equation 1.
> In Equation 1, the symbol "◦" represents the function composition. We will include detailed explanations for clarity.
>
> ### Q8: It is recommended that the authors add visible variables in the figures.
> We will label the input and output tensors' names and dimensions of each module in Figure 2 for clarity.
>
> ### Reference
> [a] Ramanishka, Vasili, et al. "Toward driving scene understanding: A dataset for learning driver behavior and causal reasoning." Proceedings of the IEEE Conference on Computer Vision and Pattern Recognition. 2018.

---

> > ### Comment · Reviewer_zhJf · 2024-08-12
> >
> > Thank you for your detailed rebuttal. While I appreciate the effort to address my concerns, I keep my score as weak accept.

---

> > > ### Author Response · Authors · 2024-08-13
> > > **Please let us know if your concerns have been addressed**
> > >
> > > Dear Reviewer zhJf,
> > >
> > > We wish to express our gratitude for your extensive review and positive comments. Your comments are invaluable to us, and **we are fully committed to thoughtfully incorporating your insights to enhance our paper**. As the discussion phase is nearing its end, we would be grateful to hear your feedback and wondered if you might still have any concerns we could address.
> > >
> > > **It would be appreciated if you could raise your score on our paper if we address all your concerns.** We thank you again for your effort in reviewing our paper.
> > >
> > > Best regards,
> > >
> > > Authors of Paper #1178

---

### Official Review · Reviewer_1BHH · 2024-07-11

**Soundness:** 2
**Presentation:** 2
**Contribution:** 3
**Rating:** 6
**Confidence:** 3

**Summary:**

1. The authors have proposed a new method for Online Open Vocabulary action detection by leveraging pretrained vision langugae models.

2. To that end they introduce 2 main modules, a distant neighboring frame transformer and an object centric Action clustering unit.

3. They train their model with three objectives on filtered versions of the Activity-Net v1.3 and Intern Vid-10M-FLT datasets.

4. They evaluate the model zero shot on the validation splits of Thumos’14 and TVseries datasets and with a base-to-novel formulation on the Thumos’14 dataset. The authors show that in the case of the former their model beats naive CLIP baselines by ~2-7 % on Thumos’14 and ~2% on TVSeries dataset and in the case of the latter beat the OadTR model trained on the seen split of Thumos’14 by ~6-9%

5. The authors also demonstrate superior inference speed compared to OadTR.

**Strengths:**

1. The authors have proposed a new method for Online Open Vocabulary action detection by leveraging pretrained vision langugae models.

2. The authors introduce 2 main modules, a distant neighboring frame transformer and an object centric Action clustering unit.

3. The authors evaluate the model zero shot on the validation splits of Thumos’14 and TVseries datasets and with a base-to-novel formulation on the Thumos’14 dataset. The authors show that in the case of the former their model beats naive CLIP baselines by ~2-7 % on Thumos’14 and ~2% on TVSeries dataset and in the case of the latter beat the OadTR model trained on the seen split of Thumos’14 by ~6-9%

4. The authors also demonstrate superior inference speed compared to OadTR

**Weaknesses:**

1. The authors claim in lines 41 and 42 that they do not use any frame-level annotations. However the Actitvity Net dataset contains annottaions for start and end times(therefore frames) for actions. And it seems that for the L_current objective in section 3.3, the text supervision is provided to the current frame. These claims seem contradictory and more clarity about this will be better.

2. In the zero shot baseline comparison the improvement in case of Thumos-14 seem to be much larger than that of TV-series. Some explanation regarding this huge disparity is neccessary. Zero shot evaluations on more datasets can help indicate the robustness of this method to different data distributions.

3. In the zero shot baseline comparison, there seems to be a very large improvement for Thumos’14 in case of the ANet model compared to the InternVid model. The improvement is 2% for the latter while ~7% for the former.  Could this be attributed to similarity of actions, between thumos’14 and Anet ? Some investigation regarding this could shed light on the previous point. if that is the case then the improvement range disparity for TV-series( ~2%) could be explained.

4. The authors have compared their results for the open vocab evaluation with only the OadTR model. Comparison with more/better Online Action detection models (like GATEhub, MiniROAD, LSTR, Colar) is necessary for a holistic evaluation of the proposed model.

5. The ablations do not contain the different parts of the action clustering unit. Some results without the Object centric decoder are needed to justify its introduction. In table 5, there is no ablation for not using the final transformer encoder.  It should be added to justify its introduction as well.

6. In Figure 2. it is not clear what is being fed from the output of the object centric decoder to the final transformer encoder. It needs to be clearly mentioned in the figure for clarity.

**Questions:**

see weaknesses

**Limitations:**

yes

---

> ### Author Rebuttal · Authors · 2024-08-06
>
> Thanks for your valuable comments. Below are our responses to your concerns:
> ### Q1: The statement "do not use any frame-level annotations" may appear contradictory.
> Thank you for highlighting this inaccurate claim. We will rephrase it as follows: "avoid utilizing fine-grained temporal annotations, which involve humans labeling actions frame by frame to ensure no actions are missed."
>
> Regarding our use of ANet, the annotations consist of timestamps and corresponding actions for each temporal segment. However, the timestamps lack the ability to differentiate between consecutive frames, leading to potential mislabeling of frames with incorrect actions. Additionally, instances within ANet may sometimes omit actions or assign incorrect labels.
>
> For example, in sample *5n7NCViB5TU.mp4* from the training set lasting 121.44 seconds and depicting discus throwing by 3 athletes (referred to as Athletes 1-3), there are discrepancies. Athlete 1's action completion was marked within the time window $[22.4, 39.5]$, while ANet indicated $[24.3, 38.1]$ as the segmentation point. Similarly, for Athlete 2, the action completion time window was $[62.5, 73.1]$, with ANet lacking a corresponding segmentation label.
>
> ### Q2: Zero shot evaluations on more datasets can help indicate the robustness.
> We validate OV-OAD's zero-shot performance on EK100, showcasing a non-trivial improvement (1.3% cAP). For further details, please refer to the response To-All-Reviewers-Q2.
>
> ### Q3: Different improvement for THUMOS’14 and TVSeries.
> Please refer to the response To-All-Reviewers-Q3 for detials.
>
> ### Q4: Comparison with more online action detection models.
> Please refer to the response To-All-Reviewers-Q1 for detials.
>
> ### Q5: The ablations on the action clustering unit.
> We will ablate different parts of action clustering block soon.
> The reason behind using the object-centric decoder lies in its capacity to generate more semantically explicit segments compared to a standard transformer decoder block, owing to its differentiable clustering module.
> This superiority has been validated in [41-42].
> Nevertheless, we will compare it with the aforementioned rival in the following ablation.
>
> ### Q6: The ablation for not using the final transformer encoder.
> Removing the final transformer encoder (also means abandon $L_{contrast}$) would leads to training anomalies and performance drops. Without the textual guidance from $L_{contrast}$, the open-vocabulary capability is significantly compromised. As shown in Table 5 (rows 2 vs. 3). we see that reducing the number of layers in the final transformer encoder from 6 to 3 results in a performance loss.
>
> ### Q7: Figure 2 needs to be clearly mentioned for clarity.
> The object-centric decoder automatically associates $n$ input frame tokens using $k$ learnable grouping embeddings, ultimately outputting grouped tokens (${\cal G}' \in \mathbb{R}^{k \times d}$) which also can globally describe the video clip. We will label the tensor dimensions of the module's outputs in Figure 2.

---

> > ### Comment · Reviewer_1BHH · 2024-08-10
> >
> > Thank you for your detailed reply.
> >
> > Most of my concerns have been addressed except for Q6 in your rebuttal where you claim "As shown in Table 5 (rows 2 vs. 3). we see that reducing the number of layers in the final transformer encoder from 6 to 3 results in a performance loss."
> >
> > The performance changes by 0.1 which is too small in my opinion to demonstrate the utility of the final transformer encoder. I suspect this is the case because the authors perform average pooling (as depicted in Figure 2.) after the encoder which makes a lot of the processing in the final encoder redundant. I would also like to point out crucially that removing the final text encoder does not mean abandoning the L_{contras} loss. I suggest that the authors remove the final encoder completely and run an experiment with the same set of objectives as before to confirm this point. This is not a major issue but i think it will improve the quality of the paper.
> >
> > My rating stays unchanged.

---

> > > ### Author Response · Authors · 2024-08-11
> > > **Response to Comments**
> > >
> > > Thank you for taking the time to review our response and for offering valuable feedback. We are glad to know that most of your concerns have been resolved.
> > >
> > > Your feedback on Q6 was particularly insightful. We initially misunderstood your question and proceeded with the experiment as per your recommendation regarding the exclusion of the final transformer encoder. The data presented in the table below indicates a marginal performance decrease of around **0.6%** upon removing the final transformer encoder. Additionally, this action results in a **15%** reduction in certain training parameters, specifically in the visual encoder.
> > >
> > > We are fully committed to addressing all your concerns and will incorporate the aforementioned experiments and analyses into the final version.
> > >
> > > | $Ψ_{DNTR}$ | $Ψ_{AC}$  | THUMOS'14 (mAP%) |
> > > |:---|:----:|:----:|
> > > |4|6-2|37.5|
> > > |4|6-0|36.9|

---

> > > > ### Comment · Reviewer_1BHH · 2024-08-12
> > > >
> > > > Thank you for carefully considering my response. I am glad you found my suggestion useful. All my concerns have been addressed.

---

> > > > > ### Author Response · Authors · 2024-08-13
> > > > > **Please let us know if your concerns have been addressed**
> > > > >
> > > > > Dear Reviewer  1BHH,
> > > > >
> > > > > We wish to express our gratitude for your extensive review and positive feedback. Your feedback is invaluable to us, and **we are fully committed to thoughtfully incorporating your insights to enhance our paper**. As the discussion phase is nearing its end, we are warmly concerned whether our rebuttal addresses your concerns.
> > > > >
> > > > > **It would be appreciated if you could raise your score on our paper if we address your concerns**. We thank you again for your effort in reviewing our paper.
> > > > >
> > > > > Best regards,
> > > > >
> > > > > Authors of Paper #1178

---

> > > > > > ### Comment · Reviewer_1BHH · 2024-08-13
> > > > > >
> > > > > > Since the reviewers have addressed all my concerns and suggestions, I have now raised my rating to weak accept.

---

> > > > > > > ### Author Response · Authors · 2024-08-14
> > > > > > > **Official Comment by Authors**
> > > > > > >
> > > > > > > Dear Reviewer  1BHH,
> > > > > > >
> > > > > > > Thank you for increasing your score! Your insights have been incredibly helpful, and we are excited to incorporate the changes based on your suggestions into our paper.
> > > > > > >
> > > > > > > Once again, thank you sincerely for your support and valuable feedback!
> > > > > > >
> > > > > > > Best regards,
> > > > > > >
> > > > > > > Authors of Paper #1178

---

> > ### Author Response · Authors · 2024-08-12
> > **Ablation Results for Action Clustering Block**
> >
> > Here, we analyze the impact of an object-centered decoder compared to a standard transformer decoder unit within the action clustering block. Both are designed to bind semantically similar static frames into a group embedding. The outcomes presented below demonstrate that the object-centered decoder outperforms the standard transformer decoder in performance.
> > |Clustering Unit|THUMOS'14 (mAP%)|
> > |:-----|:-----:|
> > | Object-Centric Decoder | **37.5** |
> > | Standard Transformer Decoder | 37.0 |

---

### Author Rebuttal · Authors · 2024-08-07

# To All Reviewers
We sincerely thank each reviewer for providing constructive comments for our paper, which are very helpful to improve our paper. Below, we address the general issues raised by the reviewers.
### Q1: Comparison with more online action detection models
We conducted a comparative analysis between OV-OAD and both the fully-supervised and base-to-novel fine-tuning methods, leveraging the open-source LSTR [44] and MAT [36] frameworks. For the fully-supervised training, we adhered to the experimental configurations outlined in [36, 44], with the deviation being the utilization of solely CLIP/ViT-B extracted RGB features as inputs. Key parameters remained consistent, such as training with 20 epochs, 'adam' optimizer, a learning rate of 7e-5, among others.
For the base-to-novel fine-tuning, we followed the experimental fine-tuning setup of [20, 33], including the Adam optimizer, a learning rate of 1e−3, and the Cosine decay function for training.
| Train-Test Split  | Methods | THUMOS’14 (mAP%) |
|:--------------:|:----------:|:----------:|
| 100%-0% | OadTR-D8 | 47.4 |
| 100%-0% | LSTR | 47.7 |
| 100%-0% | MAT-D48 | **48.2**|
|100%-0%  | CLIP-I | 28.0 |
| 100%-0% | OV-OAD | 37.5 |
| 75%-25% | LSTR | 26.9 |
| 75%-25% | MAT-D48 | 25.5 |
| 75%-25% | OadTR-D8 | 33.7 |
| 75%-25% | CLIP-I | 38.6 |
| 75%-25% | OV-OAD | **44.6** |
| 50%-50% | MAT-D48 | 7.9 |
| 50%-50% | LSTR  | 9.1 |
| 50%-50% | OadTR-D48   | 9.6 |
| 50%-50% | CLIP-I | 28.6 |
| 50%-50%| OV-OAD | **35.9** |

The results are reported in the table above. The MAT-D48 indicates that MAT utilizes the ground truth of future frames (48 frames in 12 seconds) during training. We observe that the state-of-the-art MAT (ICCV23) method achieves better performance in the fully-supervised setting, but its performance is poor in the base-to-novel setting. It is evident that traditional OAD models combined with base-to-novel fine-tuning methods are not suitable for direct application to zero-shot online action detection tasks.

### Q2: Zero shot evaluations on more datasets
We evaluated OV-OAD's performance on the challenging EPIC-KITCHENS-100 (also known as EK100) dataset. This dataset comprises first-view shots and notably deviates from the ANet data distribution.
Following [a], we conducted online action detection inference on the complete test set of EK100, covering 97 verb categories.
Similar to TVSeries, we employed per-frame calibrated average precision (cAP) as the evaluation metric.
The results presented in the table below illustrate that OV-OAD enhances online action detection for first-view videos compared to CLIP.
| Methods | Arch | EK00 (cAP%) |
|:-----|:----:|:----:|
| CLIP-I | ViT/B | 40.1|
| CLIP-II | ViT/B | 39.9|
| OV-OAD | ViT/B | **41.4**|

In addition, we also chose the challenging and large OAD dataset FineAction [b], for validation. Because of resource constraints, the results are not available at the moment, and we will add a description of the testing process and experimental results of this dataset in the final version.

### Q3: Different improvement for THUMOS’14 and TVSeries.
- Between THUMOS‘14 and TVSeries on ANet

The variance in improvements between THUMOS'14 and the TV series can be attributed to the resemblances in data distribution across these two datasets and our utilization of ANet. The substantial boost of OV-OAD on THUMOS'14 can be attributed to ANet encompassing a broader array of action categories. By employing a method of close comparison in natural language, we identified 8 similar action phrases within THUMOS'14's 20 action categories, constituting **40%** of its overall categories. In the TV series dataset, we identified 9 similar action categories out of 30, equating to **30%** of its total categories.

- Between ANet and InternVid on THUMOS‘14

Similarly, OV-OAD achieves better performance on THUMOS'14 due to the fact that ANet covers a wider range of action categories than IVid. Using the same natural language tools, we compared the coverage ratios of ANet and IVid for THUMOS categories, which stood at **40%** and **15%**, respectively. To elaborate, when contrasting the IVid and THUMOS datasets, we considered 500 high-frequency verbs as the action categories for IVid. Then, we pinpointed 3 categories from them that were similar to THUMOS'14.

### Reference
[a] Damen, Dima, et al. "Scaling egocentric vision: The epic-kitchens dataset." Proceedings of the European conference on computer vision (ECCV). 2018.

[b] Liu, Yi, et al. "Fineaction: A fine-grained video dataset for temporal action localization." IEEE transactions on image processing 31 (2022): 6937-6950.

---

> ### Author Response · Authors · 2024-08-12
> **Incorporating FineAction Evaluation Results**
>
> Here, we assess the performance of OV-OAD on the large-scale video dataset, FineAction, which includes ~4,000 uncut videos categorized into **106** classes encompassing *Household Activities*, *Personal Care*, *Socializing*, *Relaxing*, *Sports*, and *Exercise*. In line with FineAction [b], we conducted online action detection inference on the complete validation set of FinaAction, utilizing the per-frame mean Average Precision (mAP) metric. The subsequent table displays the results, illustrating OV-OAD's notable enhancement compared to the baseline methods.
> |Methods|Arch|FineAction (mAP%)|
> |:----:|:----:|:----:|
> |CLIP-I|ViT/B|26.5%|
> |CLIP-II|ViT/B|27.8%|
> |OV-OAD|ViT/B|**29.2%** ($\uparrow$ 1.4%)|
>
> Finally, we commit to addressing all concerns and incorporating the mentioned experiments in the final manuscript if the paper is accepted.

---

### Author Response · Authors · 2024-08-13
**Author Rebuttal by Authors**

Dear all,

We appreciate the reviewers for their valuable feedback, acknowledging that our work presents a **"novel"** method for online open vocabulary action detection (**Reviewers 1BHH, zhJf, and Boy7**), exhibits "significant advancement" with **"state-of-the-art performance"** (**Reviewers zhJf and Boy7**), and demonstrates "superior inference speed" (**Reviewers 1BHH and wz2n**). They have recognized the "extensive experiments" and **"comprehensive ablations"** provided (**Reviewers zhJf, Boy7, and wz2n**). Additionally, the paper's addressing of a "realistic problem setting" and the inclusion of a "robust set of ablation experiments" were noted positively (**Reviewer wz2n**). We are sincerely grateful to see the positive and valuable responses to our rebuttals from all reviewers, in which they claim that their concerns have been addressed, and the additional details mentioned in the response will be  **synced** into the final manuscript.

The core innovation of our OV-OAD lies in proposing an effective pre-training framework for online open vocabulary action detection, leveraging vision-language models to overcome the limitations of existing methods in real-world applications. **This is a non-trivial paradigm in the field of online action detection, and we are among the first to  propose an open vocabulary online action detection model, trained solely on video-text pairs without manual fine-grained frame-level annotations.** Our contributions can be summarized as:
- **Open-Vocabulary Online Action Detection**: Achieved by leveraging pre-trained vision-language models and training solely on video-text pairs without manual fine-grained frame-level annotations.
- **Action Clustering Block**: Designed to aggregate frames with similar semantics using video-text correspondence within a Transformer-based model.
- **Superior Model Performance**: Surpassing existing zero-shot methods on THUMOS'14, TVSeries, EK100, and FineAction benchmarks in the novel open vocabulary online action detection setting.
- **Comprehensive Evaluations**: Providing detailed ablation studies and analyses to validate the numerous model design choices and demonstrate the limitations of Base2Novel's zero-shot online motion detection approaches.

Once again, we appreciate the supportive feedback and strongly believe that these reviews have strengthened the work.

Sincerely,

Authors of Paper #1178

---

### Decision · Program_Chairs · 2024-09-25

**Decision:**

Accept (poster)

**Comment:**

The submission proposes to leverage pre-trained video-text models for the open-vocabulary temporal action detection task. The task requires _online_ detection, where an action needs to be recognized as soon as it appears in the video stream. Evaluations are performed on THUMOS'14 and TVSeries benchmarks, where the proposed OV-OAD framework outperforms CLIP-based approaches, and comprehensive ablation experiments are provided to demonstrate the effectiveness and efficiency of the proposed approach.

All reviewers actively engaged with the authors during the rebuttal discussion. The reviewers found most of their concerns to be addressed by the rebuttal and all reviewers recommended "weak accept". The AC agrees with the reviewers that the submission would be of value to the NeurIPS research community and thus recommends acceptance. The authors are expected to incorporate the reviewers' feedback and the relevant rebuttal experiments in the camera ready version.